# Scanning Laser Ophthalmoscopy Retromode Imaging Compared to Fundus Autofluorescence in Detecting Outer Retinal Features in Central Serous Chorioretinopathy

**DOI:** 10.3390/diagnostics12112638

**Published:** 2022-10-31

**Authors:** Fabrizio Giansanti, Stefano Mercuri, Federica Serino, Tomaso Caporossi, Alfonso Savastano, Clara Rizzo, Francesco Faraldi, Stanislao Rizzo, Daniela Bacherini

**Affiliations:** 1Department of Neurosciences, Psychology, Drug Research, and Child Health, Eye Clinic, University of Florence, AOU Careggi, 50139 Florence, Italy; 2Ophthalmology Unit, Fondazione Policlinico Universitario A. Gemelli IRCSS, 00168 Rome, Italy; 3Ophthalmology Unit, Catholic University “Sacro Cuore”, 00168 Rome, Italy; 4Ophthalmic Unit, Department of Neurosciences, Biomedicine and Movement Sciences, University of Verona, 37129 Verona, Italy; 5Torino, Eye Clinic, 18152 ASL Torino, 10134 Turin, Italy; 6Consiglio Nazionale delle Ricerche, Istituto di Neuroscienze, 56124 Pisa, Italy

**Keywords:** central serous chorioretinopathy, retinal pigmented epithelium, scanning laser ophthalmoscopy retromode imaging, fundus autofluorescence, multimodal imaging

## Abstract

Central serous chorioretinopathy (CSCR) is a retinal disease characterized by a heterogeneous clinical phenotype, depending on the influence of different factors in its pathogenesis, including the presence of subretinal fluid (SRF), trophism of the retinal pigmented epithelium (RPE) and choroidal hyper-permeability. Our study has the purpose of assessing the ability of scanning laser ophthalmoscopy (SLO) retromode imaging, compared to fundus autofluorescence (FAF), to identify outer retinal features in a cohort of patients with a diagnosis of CSCR. A total of 27 eyes of 21 patients were enrolled in our study. All patients underwent full ophthalmological examination, including fundus retinography, fundus fluorescein angiography, optical coherence tomography (OCT), FAF and SLO retromode imaging. For each patient, the following features were evaluated: SRF, the presence of pigmented epithelium detachment (PED), RPE dystrophy, and RPE atrophy. RPE dystrophy was further characterized according to the appearance in FAF of iso-, hyper- and hypo-autofluorescent dystrophy. The ability to identify each feature was evaluated for FAF and SLO retromode alone, compared to a multimodal imaging approach. FAF identified SRF in 11/14 eyes (78%), PED in 14/19 (74%), RPE dystrophy with iso-autofluorescence in 0/13 (0%), hyper-autofluorescence in 18/19 (95%), hypo-autofluorescence in 20/20 (100%), and RPE atrophy in 7/7 (100%). SLO retromode imaging identified SRF in 13/14 eyes (93%), PED in 15/19 (79%), RPE dystrophy with iso-autofluorescence in 13/13 (100%), hyper-autofluorescence in 13/19 (68%), hypo-autofluorescent in 18/20 (90%), and RPE atrophy in 4/7 (57%). SLO retromode imaging is able to detect retinal and RPE changes in CSCR patients with a higher sensitivity than FAF, while it is not able to identify the depth of lesions or supply qualitative information about RPE cells’ health status, meaning that it is less specific. SLO retromode imaging may have a promising role in the assessment of patients with CSCR, but always combined with other imaging modalities such as OCT and FAF.

## 1. Introduction

Central serous chorioretinopathy (CSCR) is a macular disease affecting young to middle-aged adults characterized by serous neuroretinal detachment with or without association with retinal pigmented epithelium detachment (PED). It is characterized by an heterogenous clinical phenotype, as the accumulation of subretinal fluid (SRF) results from the complex interaction between retinal pigmented epithelium (RPE) dysfunction, choroidal hyper-permeability and possible complications caused by choroidal neovascularization (CNV) [1,2,3,4,5,6,7,8,9]. The acute form of the disease often experiences spontaneous resolution within 3–6 months, with residual retinal symptoms in a minority of cases [10,11,12]. Non-resolving or a relapsing-remitting pattern of the pathology defines the chronic form of the disease, and it is usually associated with the inability of RPE to maintain fluid homeostasis [2,3,4]. Pathogenesis is not completely understood, and many debates are still ongoing regarding the efficacy of available treatments [13,14,15,16,17,18]. Furthermore, there is still a lack of classification of CSCR based on clinical phenotype. A systematic approach based on OCT biomarkers, such as the presence of increased choroidal thickness (pachychoroid) or the presence of RPE alterations, which are known to influence final visual prognosis and to influence response to treatment, is still missing [2,3,4,19].

Multimodal imaging represents a holistic approach to better characterize outer retinal morphology and to document its alteration with pathology. Fundus autofluorescence (FAF) uses different wavelengths to highlight the presence of naturally or pathologically occurring fluorophores, mainly lipofuscin, of the ocular fundus. In CSCR, FAF is able to identify the presence of fluid and quantify the degree and extension of RPE dystrophy. According to patterns of hypo- and hyper-fluorescence, FAF provides information about the presence of SRF, the longstanding effects of SRF on underlying RPE, the passage of gravitational fluid and previous SRF reabsorption, and RPE degeneration due to disease chronicity or scar formation after complications by neovessels. Information deriving from FAF thus represents an important prognostic value for choosing the correct therapeutic approach and its expected therapeutic outcome [20,21,22,23].

Scanning laser ophthalmoscopy (SLO) retromode imaging is a technique that uses infrared wavelengths of light to penetrate deeper layers in the retina and choroid. Thanks to a laterally deviated annular aperture, it collects asymmetrically backscattered light from one direction and blocks it from other directions, creating a shadow enhancing the contrast of every change in morphology from the RPE/Bruch membrane plane, detecting with high-sensitivity alteration in the outer retinal profile [24,25,26,27,28]. It may be able to characterize RPE alterations which are not able to be seen by conventional imaging modalities [29].

The purpose of the present study is to assess the ability of SLO retromode imaging and FAF, individually, to identify outer retinal alterations (presence of SRF, RPE dystrophy) in a cohort of patients with a diagnosis of CSCR in comparison with a multimodal imaging approach, and to detect the advantages and pitfalls of each technique.

## 2. Materials and Methods

The study protocol was carried out at Careggi Teaching Hospital in Florence, Italy. The study adhered to the Declaration of Helsinki and written informed consent was obtained from all patients.

Our study is a retrospective cross-sectional case series of eyes with diagnosis of central serous chorioretinopathy (CSCR) at different stages of disease. Diagnosis of CSCR was based on a multimodal imaging approach including spectral domain optical coherence tomography (SD-OCT) and fundus fluorescein angiography (FFA). Eyes with a previous diagnosis of CSCR, successively complicated by choroidal neovascularization (CNV), were included. Exclusion criteria included the presence of media opacities which could alter image analysis. The stage of CSCR was defined based on symptom duration and the patient’s past ophthalmological history and clinical phenotype based on RPE status. Acute CSCR was defined as when the disease was diagnosed in its first occurrence, with symptom duration not exceeding 6 months and the absence of signs of disease chronicity such as photoreceptor degeneration due to fluid persistence or signs of retinal pigmented epithelium (RPE) dystrophy. The chronic form was defined by a relapse or persistence of subretinal fluid (SRF) lasting more than 6 months or the presence of signs of photoreceptor deterioration with associated RPE abnormalities.

All patients underwent full ophthalmological examination, including fundus retinography, FFA, SD-OCT, fundus autofluorescence (FAF), and SLO retromode imaging. For each patient, the following features were evaluated: the presence of SRF, the presence of pigmented epithelium detachment (PED), the degree of RPE dystrophy, and RPE atrophy. 

Dystrophy of RPE was defined as the degeneration of the RPE/Bruch membrane complex documented by SD-OCT, with mild alterations in FAF. Atrophy of RPE was defined as marked signs of degeneration or the absence of an RPE layer in SD-OCT, coupled with significant hypo-autofluorescence with FAF signaling.

RPE dystrophy was further sub-classified into iso-autofluorescent, hyper-autofluorescent and hypo-autofluorescent dystrophy. Iso-autofluorescent RPE dystrophy was defined as RPE dystrophy which could be detected by SLO retromode imaging and confirmed by corresponding RPE alteration with SD-OCT, which could not be detected by FAF as a difference from the background autofluorescence of surrounding retina. Hyper-autofluorescent dystrophy was defined as the presence of RPE abnormalities with SD-OCT with increased autofluorescence signals in FAF. Hypo-autofluorescent RPE dystrophy was defined as the presence of RPE abnormalities or RPE layer interruptions, together with mild to moderate decreased autofluorescence at FAF compared to surrounding unaffected retina.

Patients’ clinical features were characterized thanks to a multimodal imaging approach including all the techniques cited above. The ability to identify each feature was evaluated in each patient for FAF and SLO retromode alone, and compared to a multimodal imaging approach. All images were independently reviewed by two experienced examiners (SM and DB). When not in accordance, a third examiner (FG) reviewed the images for correct characterization.

The presence of outer retinal features with multimodal imaging was expressed as a ratio between the number of eyes in which the feature could be found and the total number of eyes, and expressed as a percentage (%). The ability of individual techniques was calculated as the ratio between the number of eyes where it was possible to identify the feature in the image and the number of eyes in which the feature was identified with multimodal imaging, and expressed as a percentage (%), defining the accordance between the single technique and multimodal imaging, set as the gold standard for outer retinal features identification.

## 3. Results

Twenty-seven eyes of 21 patients (14 males) with a diagnosis of central serous chorioretinopathy (CSCR) at different stages of the disease were enrolled in our study. The mean age was 50.7 ± 5.0 years and the mean best corrected visual acuity (BCVA) was 0.26 ± 0.17 LogMAR. Overall, 4 eyes had acute CSCR and 23 patients had chronic CSCR, among which 5 were complicated by choroidal neovascularizations (CNV).

Outer retinal features detected in our cohort of patients are depicted in Table 1. In our cohort of patients, multimodal imaging identified the presence of subretinal fluid (SRF) in 14/27 eyes (52%), the presence of pigmented epithelium detachment (PED) in 19/27 (70%), retinal pigmented epithelium (RPE) dystrophy with iso-autofluorescence in 13/27 (48%), RPE dystrophy with hyper-autofluorescence in 19/27 (70%), RPE dystrophy with hypo-autofluorescence in 20/27 (74%), and RPE atrophy in 7/27 (26%).

Compared to multimodal imaging’s diagnostic ability, fundus auto-fluorescence (FAF) identified SRF in 11/14 eyes (78%), PED in 14/19 (74%), RPE dystrophy with iso-autofluorescence in 0/13 (0%), RPE dystrophy with hyper-autofluorescence in 18/19 (95%), RPE dystrophy with hypo-autofluorescence in 20/20 (100%), and RPE atrophy in 7/7 (100%), as shown in Table 2.

SLO retromode imaging identified SRF in 13/14 eyes (93%), PED in 15/19 (79%), RPE dystrophy with iso-autofluorescence in 13/13 (100%), RPE dystrophy with hyper-autofluorescence in 13/19 (68%), RPE dystrophy with hypo-autofluorescent in 18/20 (90%), and RPE atrophy in 4/7 (57%), as shown in Table 3.

## 4. Discussion

Central serous chorioretinopathy (CSCR) is a complex disease resulting from the interaction of different morphological and functional retinal and choroidal components. Its pathogenesis may derive from one of these components or the interaction between them, with RPE dysfunction, choroidal hyper-permeability, and endocrine and autonomic dysfunction among them [1,2,3,4,5,6]. In its complexity and heterogeneity, it becomes particularly important to morphologically characterize every patient to study with precision RPE integrity, choroidal vascular permeability and fluid behavior, to highlight the interaction among different pathogenetic components at the base of the clinical phenotype [19]. In our study, we wanted to show that multimodal imaging must be at the basis of disease characterization in CSCR, as single techniques are not able to replicate patient characterization accuracy as all instrumentation used synergistically. We wanted also to quantify the diagnostic performance of each technique by comparing ability to characterize retinal and choroidal features, with a focus on FAF and SLO retromode.

Fundus autofluorescence defines the metabolic status of RPE, resulting in hyper-autofluorescence in the case of fluorophore accumulation inside the cell or their clumping in the context of dystrophy, or in hypo-autofluorescence when RPE cells lose their function and do not produce and degrade any more naturally developing fluorophores, evidencing a damaged or absent RPE layer [20,21,22,23]. Various studies have highlighted the role of FAF imaging and changes over time in patients with CSCR according to the phase of the disease [22,23]. FAF patterns are various, and an increased or decreased FAF signal depends on RPE status and the presence of SRF. In our study, FAF had the best performance in characterizing RPE dystrophy, especially in the context of a mild to high degree of atrophy, while it could not detect RPE mottling which did not cause a change in fluorescence, which instead scanning laser ophthalmoscopy retromode imaging could identify. This technique was not able to detect RPE dystrophy when RPE irregularity was too mild to elicit a change in fluorescence (iso-autofluorescent RPE dystrophy) or when the presence of SRF created a blockage effect of RPE autofluorescence, or resulted in increased autofluorescence, as seen in Figure 1 and Figure 2, blocking the possibility to spot a topographical correspondence between RPE autofluorescence and OCT.

SLO retromode imaging is able to detect small retinal and RPE changes deviating from the physiological RPE/Bruch membrane level with higher sensitivity compared to FAF, spotting alterations which do not include changes in autofluorescence, or that are too small to be detected by standard retinography and are not easy to see with OCT, especially when not scanned by standard macular scans. Furthermore, thanks to retro-illumination, SLO retromode detects these RPE changes despite the presence of SRF. On the other hand, SLO retromode imaging is not able to identify the depth of lesions and to supply qualitative information about RPE cells health status, resulting in less specificity concerning tissue trophism and pigments accumulation. SLO retromode could detect RPE mottling that could not be spotted by fundus autofluorescence in 13 eyes, then confirmed by optical coherence tomography. SLO retromode had difficulty spotting RPE dystrophy that did not cause elevation above the RPE level, such as thatwith a certain degree of atrophy compared to mottling. When there are no changes in elevation above the RPE plane, as in the case of a mild to high degree of atrophy, SLO retromode has a reduced diagnostic ability compared to FAF or OCT, as seen in Figure 3 and Figure 4.

Our study has several limitations, including the retrospective design of this case series and the small sample size of the study group. 

Our study aims to characterize with increased accuracy patients with acute and chronic CSR, as the simultaneous use of SD-OCT, FAF, color fundus images and SLO retromode imaging allows us to highlight damages at the level of RPE which may be undetected by routine imaging. While FAF is a technique used routinely in the medical retina clinic, SLO retromode imaging is a novel imaging technique and is still not adopted on a large scale.

Fundus autofluorescence represents the most sensitive tool to assess functional RPE status. SLO retromode imaging may have a promising role in the assessment of patients with CSCR, but is always combined with other imaging modalities such as OCT and FAF in the context of multimodal imaging.

## Figures and Tables

**Figure 1 diagnostics-12-02638-f001:**
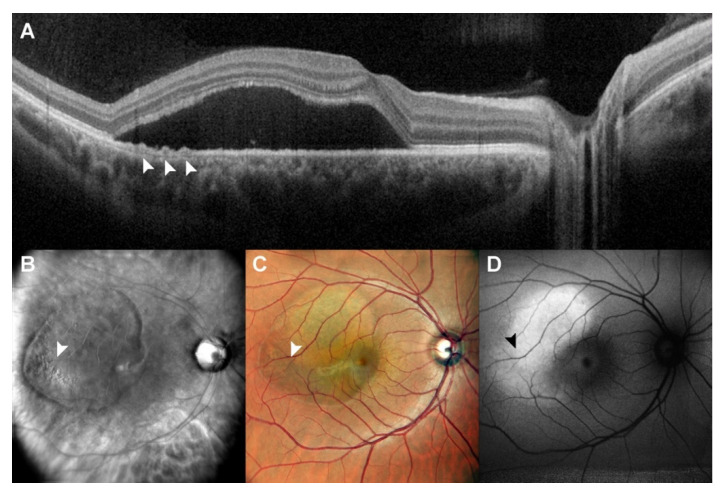
Patient with central serous chorioretinopathy assesed by multimodal imaging (I). A patient with central serous chorioretinopathy is assessed by spectral domain optical coherence tomography (SD-OCT) (**A**), scanning laser ophthalmoscopy (SLO) retromode imaging (**B**), SLO color fundus imaging (**C**) and fundus autofluorescence (FAF) (**D**). In this case, we have an extended amout of SRF, which creates hyper-autofluorescence signalling in FAF, covering a mild RPE dystrophy temporally to fovea (arrows, highlighting RPE dystrophy sseen at SD-OCT). SD-OCT was able to identify RPE elevations, while SLO retromode was able to identify RPE dystrophy despite the presence of fluid thanks to retroillumination.

**Figure 2 diagnostics-12-02638-f002:**
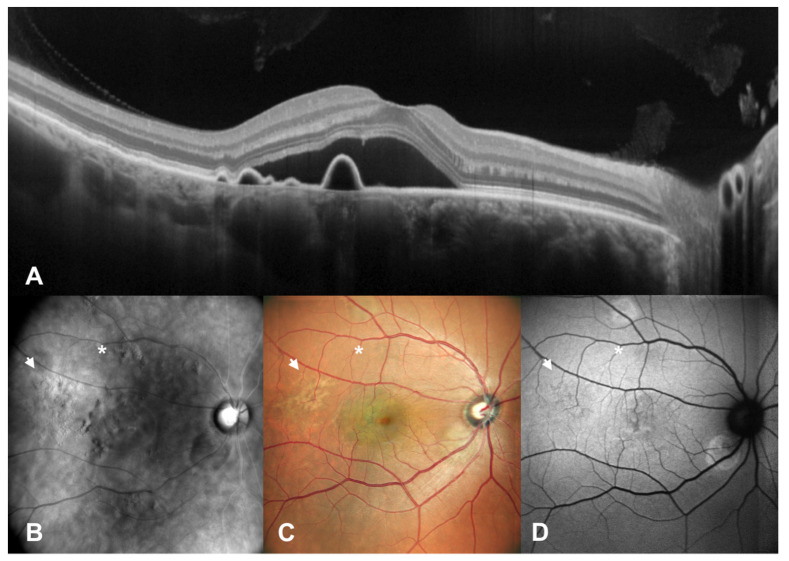
Patient with central serous chorioretinopathy assesed by multimodal imaging (II). A patient with central serous chorioretinopathy is assesed by spectral domain optical coherence tomography (SD-OCT) (**A**), scanning laser ophthalmoscopy (SLO) retromode imaging (**B**), SLO color fundus imaging (**C**) and fundus autofluorescence (FAF) (**D**). In this case, we have the presence of RPE mottling (arrows and asterisks, highlighting RPE abnormalities on fundus images), which is detectable with SLO retromode imaging, while eliciting only mild hypo-autofluorescence in FAF.

**Figure 3 diagnostics-12-02638-f003:**
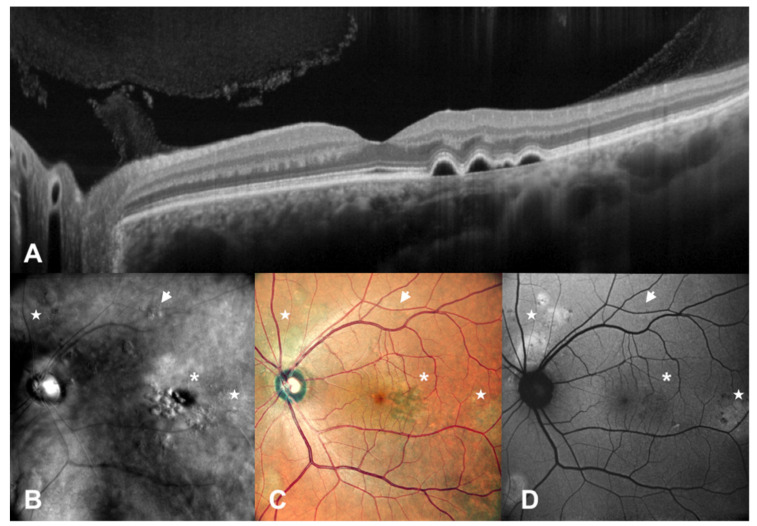
Patient with central serous chorioretinopathy assesed by multimodal imaging (III). A patient with central serous chorioretinopathy is assessed by spectral domain optical coherence tomography (SD-OCT) (**A**), scanning laser ophthalmoscopy (SLO) retromode imaging (**B**), SLO color fundus imaging (**C**) and fundus autofluorescence (FAF) (**D**). In this case, we have the presence of RPE mottling (arrows) and pigmented epithelium detachments (asterisks) which are detectable with SLO Retromode imaging, while they are not seen with FAF. Flat areas of RPE dystrophy and areas of ellipsoid zone damage, exposing RPE fluorophores, can be clearly identified with FAF, while they are difficult to spot with SLO retromode imaging. (stars).

**Figure 4 diagnostics-12-02638-f004:**
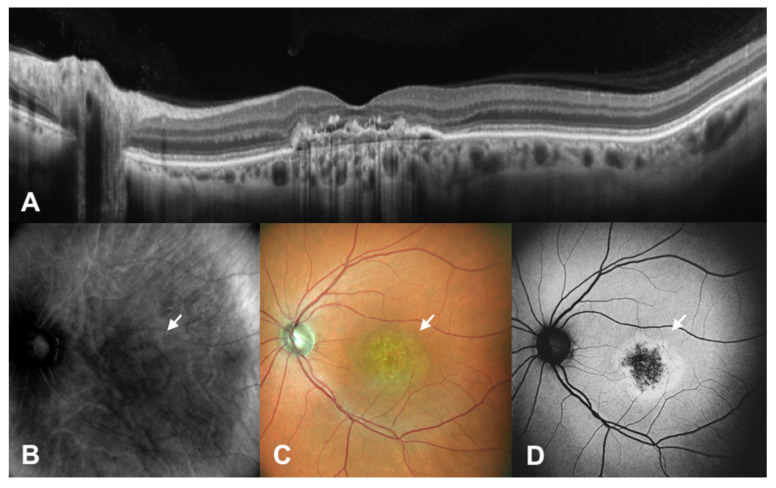
Patient with central serous chorioretinopathy assesed by multimodal imaging (IV). In this figure, we see a patient with central serous chorioretinopathy assesed by spectral domain optical coherence tomography (SD-OCT) (**A**), scanning laser ophthalmoscopy (SLO) retromode imaging (**B**), SLO color fundus imaging (**C**) and fundus autofluorescence (FAF) (**D**). In this case, we have the presence of a central RPE scar with outer retinal degeneration (arrows, highlighting atrophy at fundus images). SLO retromode imaging has difficulty in highlighting the area of RPE dystrophy, while FAF provides information on central RPE atrophy and surrounding outer retinal distruption.

**Table 1 diagnostics-12-02638-t001:** Summary of outer retinal features detected by multimodal imaging for the characterization of our study population. Outer retinal features detected in our study population. Multimodal imaging included: fundus retinography, fundus fluorescein angiography, optical coherence tomography, fundus autofluorescence and scanning laser ophthalmoscopy retromode imaging. Percentage (%) is described as the ratio between number of eyes in which the feature was detected over the total number of eyes examined.

	SRF	PED	Iso-Autofluorescent Dystrophy	Hyper-Autofluorescent Dystrophy	Hypo-Autofluorescent Dystrophy	Atrophy
**Present**	14	19	13	19	20	7
**Absent**	13	8	14	8	7	20
**%**	52%	70%	48%	70%	74%	25%

SRF = subretinal fluid; PED = pigmented epithelium detachment; % = percentage.

**Table 2 diagnostics-12-02638-t002:** Summary of outer retinal features detected by fundus autofluorescence (FAF) for the characterization of our study population. Percentage (%) is described as the ratio between the number of eyes in which the feature was detected over the total number of eyes examined. Effective % describes the ratio between the number of eyes in which FAF was able to identify the feature and the number of eyes in which multimodal imaging could detect the feature, thus the % of accordance between FAF and multimodal imaging accuracy in identifying the feature.

	SRF	PED	Iso-Autofluorescent Dystrophy	Hyper-Autofluorescent Dystrophy	Hypo-Autofluorescent Dystrophy	Atrophy
**Present**	11	14	0	18	20	7
**Absent**	16	13	27	9	7	20
**%**	40%	51%	0%	66%	74%	25%
**effective %**	78%	73%	0%	94%	100%	100%

SRF= subretinal fluid; PED= pigmented epithelium detachment; % = percentage.

**Table 3 diagnostics-12-02638-t003:** Summary of outer retinal features detected by scanning laser ophthalmoscopy (SLO) retromode imaging for the characterization of our study population. Percentage (%) is described as the ratio between the number of eyes in which the feature was detected over the total number of eyes examined. Effective % describes the ratio between the number of eyes in which SLO retromode was able to identify the feature and the number of eyes in which multimodal imaging could detect the feature, thus the % of accordance between FAF and multimodal imaging accuracy in identifying the feature.

	SRF	PED	Iso-Autofluorescent Dystrophy	Hyper-Autofluorescent Dystrophy	Hypo-Autofluorescent Dystrophy	Atrophy
**Present**	13	15	13	13	18	4
**Absent**	14	12	14	14	9	23
**%**	48%	55%	48%	48%	66%	14%
**effective %**	92%	78%	100%	68%	90%	57%

SRF = subretinal fluid; PED = pigmented epithelium detachment; % = percentage.

## Data Availability

The data presented in this study are available on request from the corresponding author. The data (original imaging) are not publicly available due to privacy issues.

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
