# Peer review of "Scanning Laser Ophthalmoscopy Retromode Imaging Compared to Fundus Autofluorescence in Detecting Outer Retinal Features in Central Serous Chorioretinopathy"

_diagnostics, 2022, doi:10.3390/diagnostics12112638_

Round 1

Reviewer 1 Report

Thank you for introducing a new valuable diagnostic method for CSCR.

[Introduction]

-       Please give a more brief overview of CSC (especially second paragraph), and describe in detail how FAF helps in approaching CSC patients and the weakness of FAF.

[Materials and Methods]

-       As stated in the introduction, there is still no standard classification system for CSC. Please reference the sentence defining the stage and be more specific.

-       What is the definition of RPE dystrophy?

-       In order to see the diagnostic ability of the test equipment, it is important who made the diagnosis by what criteria. In addition, it is necessary that at least two examiners, not one examiner, read the results and confirm the degree of agreement between the two persons.

[Results]

-       Please show more cases besides Figure 1. (iso-autoF RPE dystrophy case, hypo-auto RPE dystrophy, atrophy etc)

[discussion]

-       4th paragraph : it would be good to move on to the paragraph that describes the FAF.

[etc]

-       Caption of Table2 : hgh -> high

-       Caption of Table3 : aa -> a

Author Response

Dear reviewer, 
Thanks for your revisions which are on point. I will answer to every point in your comments.

[Introduction]

-       Please give a more brief overview of CSC (especially second paragraph), and describe in detail how FAF helps in approaching CSC patients and the weakness of FAF.

Thank you. I provided changes and further explanation on difficulty to provide a classification of CSCR despite presence of many possible phenotypes. I also provided a deeper introduction concerning FAF patterns in CSCR, as described by Han et al. (Retina 2020)  

[Materials and Methods]

-       As stated in the introduction, there is still no standard classification system for CSC. Please reference the sentence defining the stage and be more specific.

-       What is the definition of RPE dystrophy?

-       In order to see the diagnostic ability of the test equipment, it is important who made the diagnosis by what criteria. In addition, it is necessary that at least two examiners, not one examiner, read the results and confirm the degree of agreement between the two persons.

Thank you. We added in this section everything the reviewer asked and provided the requested changes.

[Results]

  •       Please show more cases besides Figure 1. (iso-autoF RPE dystrophy case, hypo-auto RPE dystrophy, atrophy etc)

Thank you. We added 3 additional figures with explanation. Our aim was to provide an insight of virtues and pitfalls of each technique.

[discussion]

  •       4th paragraph : it would be good to move on to the paragraph that describes the FAF.

Thank you. We added in this section everything the reviewer asked and provided the requested changes.

[etc]

-       Caption of Table2 : hgh -> high

  •       Caption of Table3 : aa -> a

Thank you. We added in this section everything the reviewer asked and provided the requested changes.

Reviewer 2 Report

Dear Author(s),

Thanks for your submission on Diagnostics. I have read with interest the data from your paper in which a comparative analysis between standard multimodal imaging, FAF and SLO retrocede imaging. I am convinced of the potential use of this latter technique in the retinal imaging and new works are needed to strenghten our knowledges on its pros and cons. This is a well-written paper and the comparison between the two methodics is clear and confirmed by the images provided.

Only minor issues need to be addressed:

Abstract, line 34: please correct in CSCR;

Introduction, page 2, line 46: the line "It is characterised...(CNV)" lacks of clarity, please try to amend it;

Line 66: please use "one" instead of "1";

There is an eccessive use of capital letters in all the body text. Please remove it when it is not necessary: Bruch's membrane, multimodal imaging, etc...;

Material and Methods, line 81: please replace "original" with "previous";

Line 82: please add a comma after "(CNV)";

It is not necessary to explain an acronym a second time, when you have already done it in a previous sentence. Please correct in the entire text body (i.e. line 88 FAF);

Page 3, line 96:  the sentence "Hyper-autofluorescence...unaffected retina" results unclear, please revise;

Line 106: sentence "Ability of...identification" is too long and complex (6 lines!). Please try to use two sentences to make the concept clearer;

Table and figure legend are needed to describe the findings observed in a table or a figure, not to give explanation to them (i.e. line 145 sentence "FAF had...identify"; line 161). Explanations must be inserted in the discussion paragraph;

Discussion, page 5, line 168: please add "the" before "interaction";

Line 172: the sentence "Thus, it becomes...phenotype" is not formally clear, please revise;

Line 186: The sentence "This technique...and OCT" is too long and appears difficult to be read. Please revise;

Page 6, line 207: please add "the" before the word "depth";

Line 212: You stated that various studies on FAF and CSCR are found in literature, but what utility has this sentence at the end of the discussion? If you want to discuss their results comparing to yours, please extend this part. Otherwise, this sentence must be places in the first part of the discussion;

Line 221: please add "still" before "not".

Author Response

Dear reviewer,
Thank you for the kind words of appreciation and your comments which we welcomed. Here we provide the list of the changes we made to the manuscript body.

Only minor issues need to be addressed:

Abstract, line 34: please correct in CSCR;

Introduction, page 2, line 46: the line "It is characterised...(CNV)" lacks of clarity, please try to amend it;

Line 66: please use "one" instead of "1";

There is an eccessive use of capital letters in all the body text. Please remove it when it is not necessary: Bruch's membrane, multimodal imaging, etc...;

Material and Methods, line 81: please replace "original" with "previous";

Line 82: please add a comma after "(CNV)";

It is not necessary to explain an acronym a second time, when you have already done it in a previous sentence. Please correct in the entire text body (i.e. line 88 FAF);

Page 3, line 96:  the sentence "Hyper-autofluorescence...unaffected retina" results unclear, please revise;

Line 106: sentence "Ability of...identification" is too long and complex (6 lines!). Please try to use two sentences to make the concept clearer;

Table and figure legend are needed to describe the findings observed in a table or a figure, not to give explanation to them (i.e. line 145 sentence "FAF had...identify"; line 161). Explanations must be inserted in the discussion paragraph;

Discussion, page 5, line 168: please add "the" before "interaction";

Line 172: the sentence "Thus, it becomes...phenotype" is not formally clear, please revise;

Line 186: The sentence "This technique...and OCT" is too long and appears difficult to be read. Please revise;

Page 6, line 207: please add "the" before the word "depth";

Line 212: You stated that various studies on FAF and CSCR are found in literature, but what utility has this sentence at the end of the discussion? If you want to discuss their results comparing to yours, please extend this part. Otherwise, this sentence must be places in the first part of the discussion;

Line 221: please add "still" before "not".

We provided each change you suggested in your review.

Reviewer 3 Report

The article assesses the ability of SLO Retromode imaging and FAF, to identify outer retinal alterations (presence of SRF, RPE dystrophy) in a cohort of patients with diagnosis of CSCR in comparison with a Multimodal Imaging Approach, and to detect pearls and pitfalls of each technique.

Here are somr comments:

1: The title should be a little bit shorter.

2: In introduction, too many references are placed in one sentence. We suggest the author to add some discriptions.

3: On line 87 :“All patients underwent full ophthalmological examination, including fundus retinography, SD-OCT, fundus autofluorescence (FAF), FFA, and SLO Retromode imaging, but 24 lines of the abstract All patients underwent full ophthalmological examination, including fundus retinography, optical coherence tomography (OCT), FAF and SLO Retromode imaging”, no FFA was involved.

4: On line 127, “Multimodal imaging included: fundus retinography, fundus fluorescein angiography, optical coherence tomography, fundus autofluorescence and scanning laser ophthalmoscopy retromode imaging. It is recommended that the order of these methods be the same as in the summary.

5: On the line 121, “Retinal Pigmented Epithelium (RPE) dystrophy with iso-autofluorescence in 13/17 (48%),” “13/17” should be changed to “13/27”.

6: The first line of line 180 is not indented.

7: The article assesses the ability of several methods. How about Specificity and Sensitivity of these methods? Here should be more discussion.

Author Response

Dear reviewer,
Thank you for your comments which are on point. We provided here all changes made to manuscript after reviewers comments. 

Here are some comments:

1: The title should be a little bit shorter.

We changed the previous manuscript title into " Scanning Laser Ophthalmoscopy Retromode Imaging compared to Fundus Autofluorescence in detecting outer retinal features in Central Serous Chorioretinopathy"

2: In introduction, too many references are placed in one sentence. We suggest the author to add some discriptions.

 Thank you for your comment. We increased information to enrich introduction section, adding insights on FAF and SLO retromode imaging.  

3: On line 87 :“All patients underwent full ophthalmological examination, including fundus retinography, SD-OCT, fundus autofluorescence (FAF), FFA, and SLO Retromode imaging, but 24 lines of the abstract All patients underwent full ophthalmological examination, including fundus retinography, optical coherence tomography (OCT), FAF and SLO Retromode imaging”,no FFA was involved.

 Thank you, we added FFA in the abstract section. 

4: On line 127, “Multimodal imaging included: fundus retinography, fundus fluorescein angiography, optical coherence tomography, fundus autofluorescence and scanning laser ophthalmoscopy retromode imaging. It is recommended that the order of these methods be the same as in the summary.

 Thank you. We provided the suggested changes.

5: On the line 121, “Retinal Pigmented Epithelium (RPE) dystrophy with iso-autofluorescence in 13/17 (48%),” “13/17” should be changed to “13/27”.

 Thank you. We provided the suggested change, as the was a typing error.

6: The first line of line 180 is not indented.

 Thank you. We provided the suggested change. 

7: The article assesses the ability of several methods. How about Specificity and Sensitivity of these methods? Here should be more discussion.

Thank you for your comments. We provided results on the ability of each technique to identify outer retinal features in CSCR. We provided data on the effective % of correct identification of each technique compared to multimodal imaging approach. Absolute sensitivity and specificity are difficult to determine, as multimodal imaging is precise, but in some cases may be imperfect, and because resolution of imaging is limited to the technologies present to date. Please suggest us more points to discuss if needed. We enriched the section about pearls and pitfalls of each technique.

Reviewer 4 Report

The authors have reported a novel imaging modality in the evaluation of a relatively common disease entity CSCR. While SLO Retromode was found to have increased sensitivity to identify RPE dystrophy with iso-autofluorescence compared with FAF, however it was not able to identify depth of lesions and to provide qualitative (functional) information about RPE cells. On the contrary, FAF which had the highest performance in characterizing RPE dystrophy, could not identify RPE dystrophy when the RPE irregularity was too mild or when SRF blocked the RPE autofluorescence.

Though the findings of this novel imaging modality are interesting, there are certain lacunae in this study:

1. It"s a retrospective study with small sample size.

2. The study is not conclusive about the exact role of SLO retromode imaging in the evaluation of CSCR barring a few cases of mild RPE mottling, and the authors conclude that it will be useful only in combination with multimodal imaging.

3. In multimodal imaging other modalities like FFA, ICG and OCT Angiography were not contrasted and correlated with the new modality of SLO Retromode imaging. That comparison would have been more beneficial and revealing about the role of this novel modality.

4. As this is an imaging based paper, more figures including all the multimodal imaging modalities could have been included to make it a more comprehensive study and more convincing regarding the exact role of SLO Retromode imaging in CSCR.

Author Response

Dear reviewer,
Thank you for your comments. We provided here changes according to your review. 

Though the findings of this novel imaging modality are interesting, there are certain lacunae in this study:

  1. It"s a retrospective study with small sample size.

    Thank you. We are aware of this limitation, describing it in the limitation section.  

2. The study is not conclusive about the exact role of SLO retromode imaging in the evaluation of CSCR barring a few cases of mild RPE mottling, and the authors conclude that it will be useful only in combination with multimodal imaging.

Thank you for your comments. We enriched the section about pitfalls of each technique we described. We described that SLO retromode imaging should be and cannot be used alone, as it has optimal diagnostic ability only in conjunction with OCT and FAF, to correctly describe the image provided by SLO retromode. Therefore, SLO retromode and FAF cannot be used alone in disease characterization, but add important information.  

3. In multimodal imaging other modalities like FFA, ICG and OCT Angiography were not contrasted and correlated with the new modality of SLO Retromode imaging. That comparison would have been more beneficial and revealing about the role of this novel modality.

Thank you for your comment. Diagnosis was already confirmed in many patient so we could not directly perform all exams on each patient. Our article focused on ability of SLO retromode imaging compared to FAF in detecting outer retinal features, with aa focus on RPE abnormalities. 

4. As this is an imaging based paper, more figures including all the multimodal imaging modalities could have been included to make it a more comprehensive study and more convincing regarding the exact role of SLO Retromode imaging in CSCR.

Thank you for you comment, we added 3 further images.